# Effect of Bacterial Phytase on Growth Performance, Nutrient Utilization, and Bone Mineralization in Broilers Fed Pelleted Diets

**DOI:** 10.3390/ani13091450

**Published:** 2023-04-24

**Authors:** Soudabeh Moradi, Mohammad Reza Abdollahi, Arash Moradi, Leili Jamshidi

**Affiliations:** 1Department of Animal Science, Faculty of Agriculture and Natural Resources, Razi University, Kermanshah 85438-67156, Iran; 2Monogastric Research Center, School of Agriculture and Environment, Massey University, Private Bag 11 222, Palmerston North 4442, New Zealand; 3Department of Animal Science, College of Agriculture, Ilam University, Ilam 516-69315, Iran

**Keywords:** phytase, broiler, performance, nutrient utilization, Ca to avP ratio

## Abstract

**Simple Summary:**

Knowledge of the nutrients released by microbial phytase enzymes is critical for their economic and sustainable use and precise poultry feed formulation. Several factors, including assessing measurements, age, adaptation period, phytate content in the diet, phytase dose and source, as well as calcium (Ca) to phosphorus (P) ratio, particularly in low P diets, can affect the function of phytase and magnitude of responses. The present study aimed to investigate the influence of bacterial 6-phytase on growth performance, coefficient of apparent ileal digestibility of nutrients (CAID), blood parameters, and bone mineralization in broilers that were fed a diet with a constant ratio of calcium to available phosphorus (avP). Using a constant ratio between calcium and available phosphorus led to a decrease in the calcium level of the experimental diets. The results showed that the bacterial 6-phytase dosed in the range of 500–1000 FTU/kg was effective in replacing 1.5 g/kg avP and 3 g/kg Ca in broilers fed pelleted diets, using bone quality, body weight gain (BWG) and feed per unit gain (FCR) as outcome measures. Poultry nutritionists might need to consider Ca levels in diets supplemented with phytase in feed formulations.

**Abstract:**

The influence of a bacterial 6-phytase on growth performance, coefficient of apparent ileal digestibility (CAID) of nutrients, blood parameters, and bone mineralization in broilers was evaluated. A total of 630 one-day-old male broilers were allocated to 7 dietary treatments, including positive control (PC) diet containing dicalcium phosphate, the PC marginally reduced in available P (avP) by 0.1% and calcium (Ca) by 0.2% vs. PC (NC1) or moderately reduced by 0.15 and 0.3% vs. PC (NC2), respectively, and four further diets comprising the NC1 and NC2 supplemented with 500 or 1000 FTU/kg of phytase in starter and finisher phases. A constant Ca to avP ratio was maintained across all diets. The body weight gain (BWG) and feed per unit gain (FCR) of birds fed NC1 and NC2 diets supplemented with phytase (500 and 1000 U/kg) was equivalent to that of birds fed the PC diet at 35 days. Phytase supplementation in the NC1 diet linearly increased the CAID of nitrogen (N) (*p* < 0.01), phosphorus (P) (*p* < 0.01), and Ca (*p* < 0.05). Additionally, phytase reduced (*p* < 0.01) excreta P concentration by approximately 27%, improved (*p* < 0.001) toe ash, and tended to increase tibia ash (*p* = 0.08), comparable with the PC. In conclusion, the addition of bacterial 6-phytase dosed in the range of 500–1000 FTU/kg was effective in replacing 1.5 g/kg avP and 3 g/kg Ca in broilers fed pelleted diets, using bone quality, BWG, and FCR as outcome measures.

## 1. Introduction

Microbial phytases, with the ability to hydrolyze phytate complexes well established in the poultry industry as the primary nutritional approach to improve phytate degradation. The effectiveness of phytase in improving phosphorus (P) and calcium (Ca) bioavailability, performance, and bone ash content are generally accepted, and several publications provide clear documents of these effects [1,2,3,4]. Several factors affect the function of phytase and the magnitude of responses, including assessing measurements, age of birds, adaptation period, phytate content in the diet, phytase dose and source [2], as well as Ca to P ratio, particularly in low P diets [5,6]. Studies showed that phytase enzyme has extra-phosphoric effects, which arise from the liberation of myo-inositol from phytate into the gastrointestinal tract [7,8].

Calcium is the most abundant mineral in broiler diets (about 9–10 g/kg); therefore, a large proportion of the phytate will be bound to Ca in the gastrointestinal tract, and insoluble complexes are formed that are resistant to digestion by phytase [9] and decreases the availability of both phytate phosphorus and Ca to the bird. The absolute levels and the ratio of Ca to P in the diet are considered to be critical factors influencing the digestibility, performance of birds, and exogenous phytase efficiency [10,11]. Elevated dietary Ca content can negatively affect the efficacy of exogenous phytase [12] and the utilization of phosphorus [13], so maintaining an optimal balance between Ca and P in the diet for optimal growth and bone mineralization is important [5,6]. In this context, the determination of an efficient dose level of phytase according to the ratio of calcium to phosphorus is of high relevance. Most of the studies performed on phytase enzyme were in a limited duration or after 2 weeks, as Ravindran and Abdollahi [14] revealed that evaluating broiler responses to phytase in young birds (d 0 to 10) is of particular importance because of rapid growth and development of bones and other tissues and high requirements of minerals. Although the influence of the phytase enzyme on growth performance and bone mineralization in broilers has been investigated in a number of studies, information on the optimal ratio of dietary Ca:P in the context of phytase supplementation at various dose levels is limited. Dersjant-Li et al. [11] showed that moderate (1.6 g/kg) to high (2.3 g/kg) reduction in dietary Ca in diets supplemented with 500–1000 FTU/kg of phytase improved P and Ca digestibility and growth performance. Understanding the effects of phytase enzyme on Ca and P utilization will provide valuable information that helps to better maintain Ca to P balance in broiler diets and reduce the negative impact of high levels of Ca, which can ultimately lead to improved performance.

The primary objective of the present experiment was to evaluate the effect of reducing dietary Ca content (relative to phosphorus) due to maintaining the Ca to available phosphorus (avP) ratio at a constant level in the presence of bacterial-phytase at two dose levels on performance, and nutrient utilization responses of broilers in pelleted diets from d 1 to 35. Effects on toe and tibia minerals, serum parameters, and P excretion were also checked to provide more proof of the capacity of the tested phytase to retrieve for avP and significantly deficient in Ca.

## 2. Materials and Methods

### 2.1. Diets and Experimental Treatments

Experimental procedures were conducted following the Razi University Animal Ethics Committee guidelines (IR.RAZI.REC.1400.019). The positive control diet was based on corn and soybean meal and formulated to meet minimum requirements for nutrients (adequate in P and Ca) of the birds during the starter (days 1–21) and finisher (days 22–35) phases [15]. Two negative control diets were formulated with reductions of 2.0 g/kg Ca and 1 g/kg avP (low deficiency diet, negative control 1, NC1) and 3.0 g/kg Ca and 1.5 g/kg avP (moderate deficiency diet, negative control 2, NC2) in starter and finisher phase diets, and NC1 and NC2 supplemented with 500 (NC1–500, and NC2–500) or 1000 (NC1–1000, and NC2–1000) FTU/kg of microbial phytase.

The composition and calculated analysis of the starter and finisher diets are presented in Table 1. The phytase was premixed with ground corn before mixing with the main batch in a horizontal mixer to warrant the homogenous distribution of the enzyme. Diets were fed in a pellet form. All diets were steam-conditioned at 85 °C for the 30 s in a super conditioner (Stolz, Paris, France) and then pelleted using a pellet mill (Model: 580–140, ASIAB Industrial Group, Tehran, Iran) equipped with the die ring with 3-mm holes (starter phase) and 4-mm holes (finisher phase).

The phytase (Endo-Phos^®^, Pathway Intermediates, Seoul, Republic of Korea) was a 6-phytase product produced from *E. coli* with an expected activity of 10,000 FTU/g. Titanium dioxide (TiO_2_, VWR international bvba, Leuven, Belgium) was incorporated into the finisher diet as an indigestible marker at a rate of 5 g/kg for the ileal nutrient digestibility assessment.

### 2.2. Birds and Housing

A total of 630 one-day-old male broilers (Ross 308) were supplied from a commercial hatchery on the day of the hatch. Chicks were randomized by weight and allocated to 42-floor pens (150 × 200 cm ground area, 150 cm height), each comprising 15 hatchlings.

Body weight was measured on days 1, 21, and 35 on a pen basis and used to calculate average BWG, expressed per bird basis. Feed intake (FI) was also measured for d 1 to 21, 21 to 35, and used to calculate feed per unit gain. Mortality was recorded on a daily basis. Feed per unit gain values (FCR) were corrected for the body weight of any bird that died within the experiment.

### 2.3. Determination of Coefficient of Apparent Ileal Digestibility

On day 35, three birds from each pen (a total of 18 birds per treatment), with body weight (BW) close to the average group weight, were randomly selected and euthanized by cervical dislocation and ileal digesta (lower half towards the ileocecal junction) was collected, pooled, dried, ground and stored at 4 °C until laboratory analysis. The diets and digesta samples were analyzed for dry matter (DM), ash, titanium (Ti), nitrogen (N), starch, gross energy (GE), calcium (Ca), and phosphorus (P) contents. The coefficient of apparent ileal digestibility (CAID) of nutrients was calculated using the following formula:CAID of diet component = [(Component/Ti) diet − (Component/Ti) ileum]/(Component/Ti) diet(1)
where (Component/Ti) diet = ratio of the component to Ti in the diet, and (Component/Ti) ileum = ratio of the component to Ti in the ileal digesta.

At the end of the experiment, excreta samples for two consecutive days were collected, mixed, and sub-sampled, then dried for 24 h at 100 °C and analyzed for P concentration.

### 2.4. Tibia and Toe Measurements

On d 35, the right tibia and right toe of birds (two per pen, a total of 12 birds per treatment) euthanized for ileal digesta collection were obtained and pooled per pen for further analysis. The tibia and the samples were separated from the adherent tissues and dried at 100 °C overnight. Fat was extracted from the tibia using a Soxhlet apparatus and 100% ethyl ether according to modified methods of Watson et al. [16], then transferred to a laboratory to measure Ca, P, and ash contents of the tibia and toe ash.

### 2.5. Measuring Blood Parameters

On d 35, two birds per replicate of average BW (a total of 12 birds per treatment) were randomly selected, and blood samples (3 mL) were collected from the wing vein by syringes. Serum was separated using centrifugation at 3500× *g* for 15 min and frozen at −20 °C until analysis. The alkaline phosphatase activity (ALP), total calcium, and phosphorus concentrations were measured using commercial kits (Pars Azmon, Tehran, Iran) in a clinical chemistry auto-analyzer (Mindray BS 200).

### 2.6. Chemical Analysis

The GE of diets and ileal samples was determined by adiabatic bomb colorimetry (Parr 1261, Parr Instrument Co., Moline, IL, USA) standardized with benzoic acid. Dry matter, total ash, N, and fat contents were determined according to AOAC [17]. Phosphorus and calcium contents were determined by UV spectrophotometry (method 964.06) (UV- M51, BEL engineering SRL, Monza, Italy), atomic absorption spectrometer (GBC 932 plus, GBC scientific equipment LTD, Dandenong, Australia), respectively. The concentration of Ti was determined by UV spectrophotometry according to the method of Short et al. [18]. Starch content was measured according to the procedure of Plummer, D [19] by the Anthrone method. Phytase activities were determined using the colorimetric enzymatic method (AOAC 2000.12), where one FTU (phytase unit) was defined as the quantity of enzyme that released 1 mol inorganic ortho-phosphate per minute under the conditions of the assay. The phytate content of feed samples was measured by the colorimetric procedure [20].

### 2.7. Statistical Analysis

Pen mean values were considered as the experimental unit. The experimental design was a completely randomized design with seven dietary treatments and six replicates per treatment. The general linear model procedure of the SAS software [21] was used to analyze the data. Significant differences between means were separated by the Least Significant Difference test. Significance was determined at *p* ˂ 0.05. The significance of the linear and quadratic response of increasing phytase doses and comparison between individual treatments were further performed by orthogonal polynomial contrasts.

## 3. Results

The average analyzed phytase activity was 621 and 1035 FUT/kg in the starter diet and 537 and 925 FUT/kg in the finisher diet, with target activities of 500 and 1000 FUT/kg, respectively. In general, analyzed enzyme activities in feed samples were all close to expected; only the finisher diet containing 1000 FTU was lower than expected (Table 2). The analyzed phytate content of the starter and finisher diets ranged from 0.49 to 0.55% and from 0.36–0.42, respectively (Table 1).

Phytase inclusion had a significant effect on the BWG and FCR of broiler chickens. Compared to PC, birds fed the NC diets (NC1 and NC2) exhibited reduced BW at day 21 (*p* < 0.05) and overall (*p* < 0.01) and worsened FCR during all phases (*p* < 0.01).

The growth performance of birds was improved (*p* ≤ 0.01) with phytase addition to the NC1 and NC2 diets such that it was similar to broilers fed the nutritionally adequate control diet (PC) at the starter phase (*p* < 0.05, d 1–21), finisher (*p* < 0.01, d 22–35), and overall (*p* < 0.01, d 1–35). The birds receiving the diet with supplemental 1000 FTU phytase/kg (NC2–1000) had superior (*p* < 0.01) weight gain compared with those receiving the PC during d 22–35 and 1–35 (+12.8%, and +5.6% vs. PC, respectively, *p* < 0.01). Birds fed treatments NC1–500, NC1–1000 and NC2–500 exhibited overall (day 1–35) equivalent FCR compared to PC, whereas FCR was improved in the NC2–1000 group compared with PC in the finisher and overall (by −18, and −6 points, respectively). Generally, phytase supplementation in the NC2 diet linearly (*p <* 0.01) improved BWG and FCR from hatch to day 35 (Table 3).

Compared with PC, treatment NC2–1000 during days 1–21 increased FCR (*p* < 0.01). From d 1 to 35, dietary treatments did not change the FI of broilers (Table 3, *p* ˃ 0.05).

The CAID of P was significantly reduced in birds fed the NC1–500 and NC2–500 vs. PC diets (Table 4). At a dose level of 1000 FTU/kg, phytase supplementation maintained equivalent the CAID of P in both negative control diets to PC (*p* < 0.01). Birds fed the NC1–1000 diet showed similar Ca digestibility to the PC. However, birds in treatment NC1–500, NC2–500, and NC2–1000 exhibited decreased ileal Ca digestibility compared with PC on d 35 (*p* < 0.01). The CAID of N (*p* < 0.01) and GE (*p* < 0.01) were reduced across all dietary treatments compared to the PC. Although phytase supplementation in the NC1 diet increased linearly (*p <* 0.01), the CAID of N (*p <* 0.01), Ca (*p <* 0.01), and phosphorus (*p <* 0.05). No significant effect (*p* > 0.05) of dietary phytase addition on the CAID of starch was observed.

The influence of dietary treatments on the ALP activity, Ca, and P contents of serum are shown in Table 5. No differences among treatments in ALP activity and P content of serum at day 35 were observed (*p* > 0.05). However, contrast analysis showed that phytase supplementation in the NC2 diet increased (linear, *p <* 0.01) serum P content. Phytase addition (500 and 1000 U) to NC diets increased serum Ca concentration (*p* ≤ 0.05) such that it was similar to broilers fed the PC at day 35 (*p* < 0.01).

Tibia Ca and P on day 35 were equivalent among all treatments, whereas tibia ash (*p* = 0.08) tended to be significant (Table 6). Birds fed NC1–1000, NC2–500 and NC2–1000 had numerically higher tibia ash content compared to birds fed the PC. The ash content of the toe of birds in treatments NC2–500 and NC2–1000 exceeded those of the PC. All dietary treatments, except for NC1, significantly reduced excreta P content on d 35 (*p* < 0.01, Table 6).

## 4. Discussion

The phytase recovery from the starter (500 and 1000 FTU/kg) and finisher diets (500 FTU/kg) were in close agreement with the expected activities. Only phytase recovery from finisher diets at 1000 FTU/kg was slightly lower than targeted dose levels, which may be due to sampling and measuring error and is not anticipated to have any effect on treatment outcomes. The results of phytase recoveries in pelleted diets confirm that the phytase used in this study was resistant to heat up to 85 degrees.

The current experiment focused on whether maintaining the Ca to P ratio, which results in reducing Ca content in diet with phytase supplementation, will affect the performance, bone quality, and nutrient utilization and have an additive effect on the availability of P. In the present study, there were substantial adverse effects of the feeding of deficient-P and Ca diets on the weight gain and FCR in NC groups compared to the nutritionally adequate PC-fed birds. Broilers-fed diets supplemented at either 500 or 1000 FTU/kg phytase performed equally to the birds in the PC group and appeared to compensate for Ca and P reduction in the test diets on growth performance. The similarity of all 500 FTU phytase treatments with the PC during the starter and finisher phases, in all performance measures, suggests that the phytase was able to hydrolyze enough phytate to meet P requirements to maintain growth performance during d 1 to 35 in broiler chickens.

Birds fed medium Ca and P reduction diet supplemented with 1000 FTU phytase/kg (NC2–1000) weighed 5.6% heavier (104 g) and showed better (4.1%) FCR (6 points) than those received PC diets during the 35-d trial period. Thereby, this finding is consistent with the study by Kiarie et al. [22], Dersjant-Li et al. [11], and Dersjant-Li and Kwakernaak [23], who reported equal dose-response effects of phytase (0–2000 U/kg) on performance and, or nutrient digestibility in broilers. Dersjant-Li et al. [24] reported that at a dose level of 1000 FTU/kg, while average daily gain was equivalent to PC, FCR was improved by 2.11% on d 42. Marchal et al. [4] reported that the inclusion of phytase enzyme in a diet reduced in Ca (−0.2 to −0.3% points) and total replacement of Pi at a dose level of 1000 FTU/kg maintained growth performance equivalent to a nutritionally adequate diet in all growth phases. Higher efficiency can be achieved with higher dose levels of phytase, which provides a balanced Ca:P ratio and confirms the extra-phosphoric effect of the phytase reported in previous studies [4,11,24]. A recent study by Gonzalez-Uarquin et al. [7] revealed that these extra-phosphoric effects might arise from the liberation of *myo*-inositol from phytate into the gastrointestinal tract. Increases in plasma myo-inositol [7,8,25] or increased myo-inositol content in the digestive tract, plasma and tissue of young birds and, moreover, increased myo-inositol and inositol phosphate levels in erythrocytes of hatchlings of a breeder flock when fed phytase has been reported in a study by Whitfield et al. [25].

Indeed, our findings highlighted the advantages of increasing phytase dose with providing more nutrients for young broilers.

The critical finding of this experiment is that adding 1000 U phytase to the diets with 0.3 and 0.15 percent reduction of Ca and avP (NC2) resulted in a substantial increase in weight gain and improvement in FCR to those received similar phytase levels with 1 and 2 g/kg reduction of Ca, and avP (NC1), which is probably related the less formation of insoluble Ca-phytate complex in the gastrointestinal tract, due to the lower level of Ca in the diet, as well as, lower pH of the gut, and subsequently maintaining a proper balance between Ca and P in the diet for optimal growth and bone mineralization; confirming that excessive concentrations of dietary Ca impair P availability and phytase efficacy by attaching to the active sites of phytase and by increasing gut pH in poultry diets, as well as, interactions with proteins and de novo formation of protein–phytate complexes [12,25,26,27]. Plumstead et al. [28] reported a 71% reduction of ileal phytate-P digestibility due to increasing dietary Ca from 4.7 to 11.6 g/kg in broilers. According to Selle and Ravindran [1], an acidic pH condition is required for exogenous phytase to bind to one of the reactive sites of phytate.

In fact, the magnitude of the dose-response effect of phytase can be affected by the Ca to total P ratio [11]. Moreover, the performance response and bioavailability of both Ca and P are adversely influenced by relative increases in dietary Ca or Ca:P ratio [6,29]. Phytate solubility and susceptibility to hydrolysis by the phytase enzyme in NC2 diets due to reduced gizzard pH are possibly physiological reasons for performance response in the current study. The pH and phytate degradation become even more limiting when birds are fed pelleted diets. There is evidence of relatively higher gizzard pH in birds fed pelleted diets than those fed mash diets [30]. According to Sommerfeld et al. [31], the disappearance of P up to the terminal ileum was decreased by Ca supplementation; they suggest a formation of Ca-phosphates in the small intestine due to a higher level of Ca in the diet. In the present study, there was no difference in FI of birds fed the PC compared with birds fed the NC and NC-supplemented diets. This finding is consistent with the study by Walk and Olukosi [32], who found a similar FI between the birds fed nutrient-adequate positive control (PC) and NC diets (with a reduction of Ca by 0.22%, available P by 0.20%, energy by 120 kcal/kg, and amino acids by 1 to 5%), and with NC-diets supplemented with 2000 and 4000 FTU/kg of phytase in broilers on d 28. Additionally, in the study by Walk et al. [33], FI was not influenced by the PC and NC diets supplemented with 500, 1000, and 1500 U phytase/kg.

The current study showed that reducing P content in the diet did not have any effect on the concentration of P in serum on d 35 and was in line with the ALP activity. An increase in serum ALP activity is correlated with bone disorders and may be due to Ca or NPP deficiency or an insufficient ratio of Ca to P in the diet [34,35]. The lack of serum P and ALP activity response to Ca and P deficient diets is not in agreement with the finding by Baradaran et al. [36], who reported that feeding broiler chickens with low-NPP diets (1.5 and 2.3 g NPP/kg) reduced serum P concentrations than those fed a control diet (4 g NPP/kg) at 42 d post-hatch. Additionally, they reported a quadratic decrease in serum ALP activity due to increasing phytase supplementation. The concentration of Ca and P in the blood is preserved within a restricted physiological range through feedback mechanisms, including parathyroid hormone (PTH), active vitamin D3, calcitonin, and their respective receptors localized in the small intestine, bone, and kidneys until bone reserves are severely depleted [37]. Due to these feedback mechanisms, probably the P level in the serum and, consequently, the activity of ALP did not change. In agreement with the current results, Omotoso et al. [38] revealed that the concentration of serum metabolites (P, Ca, albumin, calcidiol, PTH, and T3) did not change in depleted P groups in broilers on d 37 due to the effectiveness of physiological adaptation mechanisms to P deficiency in the late growth phase (d 37) than in the early growth phase (d 17, 24).

The NC diet resulted in reduced serum Ca concentration compared to the PC diet, which may indicate that Ca was deficient in NC diets. The phytase-supplemented diets increased serum Ca concentration to a value not different from the PC diet. However, in the study of Zhao et al. [39], the level of Ca and P in the serum did not change in response to the Ca-deficient diet in laying hens (1.5 and 3.7% Ca). Tibia ash is a valuable measure to determine P availability in Pi sources, also the capacity of phytase sources to release phosphorus for use by birds [5,40]. Almost 80% of total body P is found in bones, and phosphorus is required as part of hydroxyapatite. In the current study, tibia ash and toe ash responded to phytase enzyme and was equivalent or increased compared to PC, and was in line with performance responses, a finding which is consistent with that of Dersjant-Li et al. [13], who reported a similar bone density during the finisher stage in highly Ca-reduced diets when supplemented with phytase.

This improvement, to a major extent, contributed to the degradation of phytate by the phytase and is an indirect confirmation of whether Ca and P were in balance in the test diets [4]. Reduced tibia ash in NC diets agrees with the study of Dhandu and Angel. [41], and Yan et al. [42] reported that even where no apparent impairment of growth performance was observed in response to reduced levels of nPP or Pi in the diet, tibia ash (weight or percent) or breaking strength was reduced.

There was a clear response effect of the phytase on the CAID of P. At a dose level of 1000 FTU/kg, CAID of P was improved by 16.7% and 8.4%, vs. NC1 and NC2 diets, respectively, and suggested that phytase supplemented at 1000 FYT/kg was still beneficial for growth in the broiler chickens. The improved ileal P digestibility was clear proof of improvements in performance and tibia ash [24] and followed the same pattern. This finding is consistent with the results of the study by Woyengo and Wilson. [43], showing a similar CIAD of P in broilers fed PC and NC containing 1000 FTU/kg of phytase. Scholey et al. [44] reported there were no significant differences in performance, tibia strength and Ca and P digestibility between the control diet and the diet containing 1000 FTU phytase and low inorganic P during the grower and finisher phases. The CAID of P for the NC diet containing 500 FTU phytase/kg was lower than that for the PC diet, which was unexpected. However, growth performance and tibia ash content of broiler chicks fed PC and NC1–500 and NC2–500 diets did not differ at 35 days, implying that the efficiency of P utilization in PC diets for growth performance and bone mineralization was low. The higher P digestibility at 1000 FYT/kg relative to 500 FYT/kg is indicated the potential for enhanced P availability beyond the currently recommended phytase inclusion level in the industry.

The CAID of GE and N values for the PC diet were greater (*p* < 0.05) than those for the NC diets supplemented with 500 and 1000 U of phytase, a finding which is challenging to explain because the NC and PC diets were formulated to be similar in terms of CP, standardized ileal digestible amino acid content and ME value. A possible reason might be that dietary Ca and non-phytate P levels influence the ileal digestibility of amino acids or CP in broilers. An increase in ileal digestibility of amino acids in broilers due to higher dietary levels of Ca and non-phytate P in a study by Kiarie et al. [22] and Adedokun et al. [45] and an increase in ileal digestibility of CP in broilers due to an enhancement in the dietary level of Ca in the study by Abdollahi et al. [46] provides support to this hypothesis. Adedokun et al. [45] found that high dietary Ca increased AID for most of the amino acids in birds fed a DDGS-based diet. According to Xue et al. [47], the total tract N retention and P retention in the low-CP diet increased linearly with increasing dietary P levels (from 0.18 to 0.59%). They suggested that the relationship between N and P retention is complicated and needs further investigation.

This finding is consistent with that of Woyengo and Wilson [43], who reported a decrease in the CAID of GE and CP in the NC diet containing 1000 FTU of phytase compared to the PC diet. However, in the study of Abdollahi et al. [46], 1000 FYT/kg phytase increased the digestibility of N by 3.68%, fat by 3.53%, and GE by 3.54%.

Phytase inclusion decreased excreta P concentration by approximately 27%, which could be, to a major extent, attributed to the releasing effect of phytase enzyme on PA-bound P, and it leads us toward more environmentally sustainable broiler production. Similar to current results, Mulvenna et al. [48] reported a 20% reduction during 0–28 d on a g/kg BW gain basis and a reduction of 15% in total P excretion to 2.2 kg target weight in broilers fed phytase-supplemented diets.

Depending on phytase biochemical characteristics and its efficacy in breaking down phytate, the minimum and optimum dose level of each phytase for the reduction of Pi and Ca in the basal diet is likely to be different. This finding suggested that a dose level of 500 FTU/kg of bacterial 6-phytase in a moderate NC2 diet (with the capacity to replace 1.5 g/kg avP and 3 g/kg Ca) is sufficient to provide adequate P for bone mineralization in broilers, however, based on the digestibility results of the current study, a 1000 U/kg of phytase for proper supply of digestible Ca and P in the diet is recommended.

## 5. Conclusions

These findings highlighted that the use of excessive quantities of Ca in broiler diets to provide a safety margin could be minimized, having significant economic and environmental consequences. Applying a total Ca to available P ratio of 2 in broiler diets improves Ca:P balance leading to improved performance, bone mineralization, and production advantages. Clearly, growth and bone responses to phytase were most remarkable at the low avP and low Ca levels. Poultry nutritionists might need to consider Ca levels in diets supplemented with phytase in feed formulations. Providing safety margins for dietary Ca supply does not reveal beneficial outcomes for the birds.

## Figures and Tables

**Table 1 animals-13-01450-t001:** Composition, calculated analysis, and analyzed values (g/kg, as fed basis) of the basal diets ^1^.

	Starter Phase			Finisher Phase		
	PC	NC1 ^2^	NC2 ^3^	PC	NC1 ^2^	NC2 ^3^
Ingredients (g/kg)						
Corn	485.5	502	510	505.8	522	530.2
Soya Meal, 43%	394.4	391.2	389.8	332	328.4	327
Wheat Flour	50	50	50	100	100	100
Vegetable Oil	26.2	20.6	17.8	25	19.8	17
Dicalcium Phosphate	18.6	12.8	10	15.1	9.4	6.5
Limestone	10.6	8.7	7.7	9.4	7.5	6.6
Salt	1.9	1.9	1.9	2.1	2.1	2.1
Min. and Vit. Premix ^4^	2	2	2	2	2	2
DL-Methionine	3.3	3.2	3.2	2.8	2.8	2.8
L-Lysine HCL	2.6	2.7	2.7	2	2.1	2.1
L-Threonine	1.3	1.3	1.3	1.2	1.2	1.2
Bicarbonate Na	2.1	2.1	2.1	1.6	1.7	1.7
Choline Chloride	1.5	1.5	1.5	1	1	1
Calculated Analysis, %						
ME, kcal/kg	2900	2900	2900	2970	2970	2970
Crude Protein	22	22	22	20	20	20
Calcium	0.96	0.76	0.66	0.83	0.63	0.53
Available phosphorus	0.48	0.38	0.33	0.415	0.315	0.265
Analyzed values, %						
Dry matter	93.7	92.2	92.3	94.4	95.1	94.9
Crude protein (nitrogen × 6.25)	21.59	22.09	22.25	21.02	20.91	21.17
Gross energy, kcal/kg	4218.3	4356.3	4203.3	4323.6	4321.1	4278.9
Fat	5.4	5.03	4	4.5	4.83	4.66
Calcium	0.976	0.842	0.705	0.876	0.746	0.583
Total phosphorus	0.75	0.63	0.579	0.588	0.508	0.363
Phytate	0.55	0.54	0.49	0.42	0.4	0.36

^1^ Phytase enzyme was added on top of the NC diets. ^2^ NC1: Ca and avP were reduced by 0.2 and 0.1%, respectively, from the PC. ^3^ NC2: Ca and avP were reduced by 0.3 and 0.15%, respectively, from the PC. ^4^ Supplied the following per kilogram of diet: vitamin A, 10,000 IU; vitamin D3, 2000 IU; vitamin E, 45 IU; vitamin K3, 1.75 mg; vitamin B12, 17 μg; riboflavin, 8 mg; niacin, 56 mg; d-pantothenic acid, 15 mg; folic acid, 2.1 mg; vitamin B6, 4 mg; thiamine, 2.7 mg; d-biotin, 190 μg; antioxidant, 0.5 mg; iron, 22.5 mg; manganese, 120 mg; zinc, 113 mg; iodine, 1.33 mg; copper, 17 mg; selenium, 0.34 mg.

**Table 2 animals-13-01450-t002:** Recovered phytase ^1^ activities in the experimental diets.

Diets	Starter Phase	Finisher Phase
NC1 + 500 FTU/kg	630	519
NC1 + 1000 FTU/kg	1040	940
NC2 + 500 FTU/kg	612	555
NC2 + 1000 FTU/kg	1030	910

^1^ The phytase used was Endo-Phos (Pathway Intermediates, Republic of Korea) with an expected activity of 10,000 FTU/g.

**Table 3 animals-13-01450-t003:** Influence of phytase supplementation on broiler growth performance from day 1 to 35 post-hatch.

	Hatch to Day 21			Hatch to Day 35		
Treatments	BW Gain, g	FI, g	FCR, g:g	BW gain, g	FI, g	FCR, g:g
PC ^1^	868.8 ^a^	1083	1.23 ^c^	1836.6 ^b^	2801	1.514 ^bc^
NC1 ^2^	806.9 ^b^	1016.1	1.264 ^bc^	1729.06 ^c^	2768.4	1.602 ^a^
NC1 + 500 FTU	872.3 ^a^	1066.5	1.224 ^c^	1869.1 ^ab^	2861.3	1.53 ^bc^
NC1 + 1000 FTU	831.6 ^ab^	1025.7	1.233 ^c^	1817.9 ^b^	2810.6	1.546 ^b^
NC2 ^3^	790.7 ^b^	1031.7	1.306 ^a^	1728.9 ^c^	2782.3	1.61 ^a^
NC2 + 500 FTU	848.6 ^ab^	1071.4	1.262 ^bc^	1873.2 ^ab^	2810.3	1.5 ^c^
NC2 + 1000 FTU	848.2 ^ab^	1084.5	1.278 ^ab^	1940.4 ^a^	2818.1	1.452 ^d^
SEM ^4^	19.08	20.13	0.013	26.88	29.86	0.014
ANOVA *p*-value	0.04	0.08	0.001	˂0.0001	0.47	˂0.0001
Linear phytase, NC1 ^5^	0.18	0.33	0.07	0.02	0.24	0.02
Linear phytase, NC2 ^5^	0.009	0.02	0.07	˂0.0001	0.49	˂0.0001
Orthogonal contrast *p*-value						
PC vs. NC1	0.057	0.02	0.07	0.007	0.44	0.0001
PC vs. NC2	0.01	0.08	0.001	0.007	0.66	˂0.0001
PC vs. NC1 + 1000 FTU	0.15	0.04	0.85	0.6	0.82	0.09
PC vs. NC2 + 1000 FTU	0.44	0.95	0.01	0.009	0.69	0.004

Means in the column with the same letter are not significantly different. NS, not significant. ^1^ PC: Positive control. ^2^ NC1: Ca and avP were reduced by 0.2 and 0.1%, respectively, from the PC. ^3^ NC2: Ca and avP were reduced by 0.3 and 0.15%, respectively, from the PC. ^4^ SEM, Pooled standard error of the mean. ^5^ Excluding PC from the analysis.

**Table 4 animals-13-01450-t004:** Influence of phytase supplementation on the coefficient of apparent ileal digestibility of calcium, phosphorus, nitrogen, gross energy, and starch in broiler chickens on d 35.

Treatments	CAID of Ca	CAID of P	CAID of Ash	CAID of N	CAID of GE	CAID of Starch
PC ^1^	0.596 ^a^	0.762 ^a^	0.421 ^a^	0.786 ^a^	0.811 ^a^	0.953
NC1 ^2^	0.434 ^c^	0.616 ^b^	0.3 ^b^	0.676 ^c^	0.728 ^b^	0.947
NC1 + 500 FTU	0.504 ^bc^	0.648 ^b^	0.315 ^b^	0.753 ^b^	0.714 ^bc^	0.954
NC1 + 1000 FTU	0.539 ^ab^	0.783 ^a^	0.275 ^b^	0.749 ^b^	0.708 ^bc^	0.956
NC2 ^3^	0.478 ^bc^	0.677 ^ab^	0.287 ^b^	0.739 ^b^	0.72 ^b^	0.942
NC2 + 500 FTU	0.503 ^bc^	0.642 ^b^	0.279 ^b^	0.739 ^b^	0.695 ^bc^	0.952
NC2 + 1000 FTU	0.519 ^b^	0.761 ^a^	0.256 ^b^	0.719 ^b^	0.683 ^b^	0.954
SEM ^4^	0.011	0.033	0.02	0.011	0.011	0.004
ANOVA *p*-value	0.001	0.004	˂0.0001	˂0.0001	˂0.0001	0.32
Linear phytase, NC1 ^5^	0.02	0.006	0.47	0.005	0.52	0.23
Linear phytase, NC2 ^5^	0.38	0.07	0.48	0.18	0.4	0.2
Orthogonal contrast *p*-value						
PC vs. NC1	˂0.0001	0.002	0.0001	˂0.0001	˂0.0001	0.37
PC vs. NC2	0.002	0.04	˂0.0001	0.01	˂0.0001	0.06
PC vs. NC1 + 1000 FTU	0.1	0.65	˂0.0001	0.02	˂0.0001	0.57
PC vs. NC2 + 1000 FTU	0.02	0.98	˂0.0001	0.0002	˂0.0001	0.84

Means in the column with the same letter are not significantly different. NS, not significant. ^1^ PC: Positive control. ^2^ NC1: Ca and avP were reduced by 0.2 and 0.1%, respectively, from the PC. ^3^ NC2: Ca and avP were reduced by 0.3 and 0.15%, respectively, from the PC. ^4^ SEM, Pooled standard error of the mean. ^5^ Excluding PC from the analysis.

**Table 5 animals-13-01450-t005:** Influence of phytase inclusion on the alkaline phosphatase activity (ALP, U/L), calcium. (Ca, mg/dL), and phosphorus (*p*, mg/dL) contents of serum in broilers on d 35.

Treatments	ALP	Ca	P
PC ^1^	908.7	10.09 ^a^	7.18
NC1 ^2^	594.1	8.33 ^b^	6.5
NC1 + 500 FTU	800.6	9.16 ^ab^	6.5
NC1 + 1000 FTU	590.1	9.75 ^a^	6.58
NC2 ^3^	819.7	8.5 ^b^	5.91
NC2 + 500 FTU	762	9.6 ^a^	6.1
NC2 + 1000 FTU	690.6	9.81 ^a^	6.81
SEM ^4^	135.4	0.33	0.31
ANOVA *p*-value	0.77	0.001	0.12
Linear phytase, NC1 ^5^	0.54	0.005	0.98
Linear phytase, NC2 ^5^	0.84	0.01	0.01
Orthogonal contrast *p*-value			
PC vs. NC1	0.19	0.0002	0.13
PC vs. NC2	0.69	0.0007	0.005
PC vs. NC1 + 1000 FTU	0.17	0.47	0.18
PC vs. NC2 + 1000 FTU	0.34	0.57	0.42

Means in the column with the same letter are not significantly different. NS, not significant. ^1^ PC: Positive control. ^2^ NC1: Ca and avP were reduced by 0.2 and 0.1%, respectively, from the PC. ^3^ NC2: Ca and avP were reduced by 0.3 and 0.15%, respectively, from the PC. ^4^ SEM, Pooled standard error of the mean. ^5^ Excluding PC from the analysis.

**Table 6 animals-13-01450-t006:** The concentration of ash, calcium (Ca), and phosphorous (P) in the tibia (%, dried defatted value), toe ash content (%, dried defatted matter), and phosphorous content of excreta (%) of broiler chickens on d 35.

Treatments	Tibia Ca	Tibia P	Tibia Ash	Toe Ash	Excreta P
PC ^1^	14.74	7.8	29.84	10.09 ^b^	0.702 ^a^
NC1 ^2^	15.13	7.47	28.87	9.69 ^b^	0.75 ^a^
NC1 + 500 FTU	15.51	7.56	29.22	10.14 ^b^	0.615 ^b^
NC1 + 1000 FTU	15.89	7.55	31.46	9.99 ^b^	0.42 ^d^
NC2 ^3^	14.88	7.75	27.76	9.77 ^b^	0.442 ^d^
NC2 + 500 FTU	14.85	7.48	30.84	11.52 ^a^	0.47 ^d^
NC2 + 1000 FTU	14.74	7.35	30.23	11.25 ^a^	0.537 ^c^
SEM ^4^	0.81	1.15	1.62	0.492	0.02
ANOVA *p*-value	0.93	0.33	0.08	0.006	˂0.0001
Linear phytase, NC1 ^5^	0.76	0.98	0.04	0.7	0.0001
Linear phytase, NC2 ^5^	0.99	0.29	0.09	0.14	0.04
Orthogonal contrast *p*-value					
PC vs. NC1	0.73	0.86	0.39	0.57	0.12
PC vs. NC2	0.9	0.02	0.002	0.32	0.0001
PC vs. NC1 + 1000 FTU	0.31	0.69	0.23	0.84	0.0001
PC vs. NC2 + 1000 FTU	0.4	0.47	0.77	0.02	0.0001

Means in the column with the same letter are not significantly different. NS, not significant. ^1^ PC: Positive control. ^2^ NC1: Ca and avP were reduced by 0.2 and 0.1%, respectively, from the PC. ^3^ NC2: Ca and avP were reduced by 0.3 and 0.15%, respectively, from the PC. ^4^ SEM, Pooled standard error of the mean. ^5^ Excluding PC from the analysis.

## Data Availability

All available data are incorporated into the manuscript.

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
