# Peer review of "Effect of Bacterial Phytase on Growth Performance, Nutrient Utilization, and Bone Mineralization in Broilers Fed Pelleted Diets"

_animals, 2023, doi:10.3390/ani13091450_

Round 1

Reviewer 1 Report

Respected Editor,

Current manuscript is novel and well structured. I have reviewed the manuscript and found it interesting because Information on the optimal ratio of dietary calcium:phosphorus in the context of phytase supplementation at various dose levels is limited. Therefore, understanding the effects of phytase enzyme on low Ca and P diets and Ca and P utilization will provide valuable information that will helps to better maintain Ca to P balance in broiler diets and reduce the negative impact of a high levels of Ca.

I have few suggestions to improve the manuscript. If authors agrees to revise manuscript accordingly, please accept it

Please consider calcium and phosphorus abbreviation and term use in the whole manuscript

Please provide exact P values in table instead of NS

Line 15: what do you mean by ‘differences in assessment criteria’ please explain

Line 23: what are ‘BWG and FCR’. The terms should be abbreviated on first seen

Line 32-33: the ratio seems incorrect, please correct it ‘The Ca to avP ratio in the starter and finisher diets was 2’

Line 33: abbreviate the terms on first seen please ‘The BWG and FCR of birds’

Lie 66-67: please correct it ‘…………on growth performance and bone mineralization in broilers has been studied, Information on the optimal ratio of dietary ……’

Line 92-93: please remove the sentence ‘The ratio of calcium to avP in all diets was maintained in the starter and finisher diets at 2’

Line 93: what do you mean by chemical composition ‘chemical composition’

May I know why corn and oil level is fluctuating so much in the experimental diets? isn’t enough to change the response instead of your experimental treatments?

What is GE in table 1 please explain in foot note

Line 117-120: please remove it ‘The room temperature was controlled at 32 °C on d 1 and was gradually reduced to 22 °C by d 21. The lighting regimen of birds was 23L:1D from d 3 to 7, 18L:6D from d 8 to 21, 20L:4D from d 22 to 28, and 23L:1D from d 28 to 35. The feed was fed ad libitum, and  water was available freely’ because it is normal practice

Line 151: ‘On d 35, two birds per replicate of average BW………….’ BW is not abbreviated elsewhere. Please define term on first seen

Line 158: Gross energy has already been abbreviated, please use abbreviation ‘The gross energy of diets and ileal samples was determined by…..’

Line 188: feed per unit gain has already been abbreviated. Please use abbreviation here

Line 212: feed intake has already been abbreviated please use abbreviation ‘From d 1 to 35, dietary treatments did not change the feed intake of broilers’

Line 213: please remove the sentence ‘No significant differences in mortality rates were observed (data not shown)’

Line 224-226: please don’t use abbreviation in title ‘Influence of phytase supplementation on the coefficient of apparent ileal digestibility (CAID) of calcium (Ca), phosphorus (P), nitrogen (N), gross energy (GE), and starch in broiler chickens on d 35’

Line 267-268: rewrite the sentence ‘The results of phytase recoveries in pelleted diets confirm that the phytase used in this study is resistant to heat up to 85 degrees’ the new sentence could be ‘The results of phytase recoveries in pelleted diets confirm that the phytase used in this study was resistant to heat up to 85 degrees’

Line 286: what is ADG in the sentence ‘Dersjant-Li et al. [22] reported that at a dose level of 1000 FTU/kg, while ADG

Line 299: what is GIT in the sentence ‘soluble Ca-phytate complex in the GIT….’

Line 318-322: please rewrite justification of citing articles ‘In agreement with the current results, Walk and Olukosi, [30] found a similar FI between the birds fed positive control and NC diets supplemented with 2000 and 4000 FTU/kg of phytase in broilers on d 28. Also, in the study by Walk et al. [31], feed intake was not influenced by the PC and NC diets supplemented with 500, 1000, and 1500 U phytase/kg’

Because these both articles

30. Walk, C. L, and O. A. Olukosi. 2019. Influence of graded concentrations of phytase in high-phytate diets on growth performance, 473 apparent ileal amino acid digestibility, and phytate concentration in broilers from hatch to 28 D post-hatch. Poult. Sci. 2019, 0:1– 474 10, http://dx.doi.org/10.3382/ps/pez106. 475

31. Walk, C. L., T. R. Santos, and M. R. Bedford. Influence of superdoses of a novel microbial phytase on growth performance, tibia 476 ash, and gizzard phytate and inositol in young broilers. Poult. Sci. 2014, 93:1172–1177, http://dx.doi.org/10.3382/ps.2013-03571.  

Have nothing to do with your low calcium diet results

Line 320: feed intake has already been abbreviated please use abbreviation ‘Also, in the study by Walk et al. [31], feed intake……………..’

Author Response

Thank you for reviewing the manuscript. 

  • all abbreviations of terms on first seen was edited.

    question: May I know why corn and oil level is fluctuating so much in the experimental diets? isn’t enough to change the response instead of your experimental treatments?

    answer: Our goal in this experiment was to ensure that all the nutrients provided by the diets were equal, except for calcium and phosphorus. Therefore, with the decrease in the level of di calcium phosphate in the diet, the amount of oil also decreased.

  • Line 318-322: edited. please see line 302-307 (red font).

Because these both articles Have nothing to do with your low calcium diet results.

Answer: In both of these studies, the amount of calcium and phosphorus in the negative diets has decreased compared to the positive ones, similar to our experiment, for example in study of Walk and Olukosi, (2019), dietary treatments consisted of a nutrient-adequate positive control (PC), a negative control (NC) diet formulated with a reduction of Ca by 0.22%, available P by 0.20%, energy by 120 kcal/kg, and amino acids by 1 to 5% when relative to the PC. The NC diet was then supplemented with 2,000 or 4,000 phytase units (FTU)/kg to create a total of 4 experimental diets. And in study of Walk et al. (2014), dietary treatments were positive control (PC); PC plus dicalcium phosphate, PC plus 500 U/kg of microbial phytase, NC (reduction of Ca and avP by 0.16 and 0.15%, respectively, from the PC) and the NC plus 500, 1000, ad 1500 U/kg of phytase.

All your suggestions have been implemented in the manuscript and marked in red.

Thank you again

  •  

Reviewer 2 Report

Dear Editor

Thank you for giving me the opportunity to evaluate this manuscript entitled “Effect of Bacterial Phytase on Growth Performance, Nutrient Utilization, and Bone Mineralization in Broilers Fed Pelleted Diets”. Because evaluation the the effect of reducing dietary Ca content (relative to phosphorus) due to maintaining the Ca to avP ratio at a constant level in the presence of bacterial-phytase is necessary, which it gives a correct understanding for the farmers. In this manuscript, they stated that the use of excessive quantities of Ca in broiler diets to provide a safety margin could be minimized, having significant economic and environmental consequences. And they could also have reported that poultry nutritionists might need to consider Ca level in diets supplemented with phytase in feed formulations. So, based on my opinion the manuscript merits acceptance.

Regards

Author Response

Many thanks for review the manuscript. 

Reviewer 3 Report

The manuscript describes the results of a feeding trial on poultry fed low P and low Ca levels with/without the supplementation of phytase (500-1000 FTU/kg) throughout the entire production period from 1-35 days of life (n=630). The aim of the presented research study was to evaluate the effect of a microbial phytase on growth performance, ileal digestibility of nutrients, some related blood parameters and bone traits in broiler chickens. While the dietary regimen was applied for n=90 per dietary treatment, the specific analyses made use of subsets of animals, i.e. a reduced number of biological replicates. The feeding trial addresses environmental aspects of livestock production that are of great importance in terms of social acceptance and sustainable food production.

The aspect of myo-inositol release and related physiological effects should be mentioned in the Introduction/Discussion sections. Please stress that the application of phytases might lead to higher levels of inorganic P in litter and therefore higher loads of inorganic P in soil with potential higher runoff. Add respective novel literature in the Introduction section:

https://www.sciencedirect.com/science/article/pii/S0032579122007647

https://www.sciencedirect.com/science/article/pii/S0032579122006459

https://www.mdpi.com/2076-2615/12/13/1669

https://www.wageningenacademic.com/doi/abs/10.3920/JAAN2021.0014

A number of specific aspects require amendment:

-          Line 80: “dig P”. It is meant “digestible P”, “available P”?

-          Line 100: What about the analyzed values (wet lab) of the applied diets? It is important to state the actual values, at least for Calcium and total/available Phosphorus (similar to the overview for phytase activity in Table 2).

-          Line 127: Provide total number of animals used for the analyses, I assume it is three birds per pen, i.e. n=18 animals per treatment?

-          Line 144: Provide total number of animals used for the analyses, I assume it is two birds per pen, i.e. n=12 animals per treatment?

-          Line 151: Provide total number of animals used for the analyses, I assume it is two birds per pen, i.e. n=12 animals per treatment?

-          Line 172: Why authors used pen-based data but not the individual data with pen as an effect considered in the model? This has been done for all statistical evaluation or only for performance data?

-          Line 254: I do not understand why authors used “phosphorous content of excreta (%)” in Table 6 – would be clearer in e.g. “[g/d]”. Please explain benefits.

-          Line 374: Consider current literature on that subject, i.e., macro- and micronutrient digestibility might be interlinked (interlinkage of N and P metabolism). https://www.sciencedirect.com/science/article/pii/S003257911931884X

Round 2

Reviewer 1 Report

I’m satisfied, please go ahead

Reviewer 3 Report

The authors adressed all raised points.